# The Reliability and Sensitivity of Change of Direction Deficit and Its Association with Linear Sprint Speed in Prepubertal Male Soccer Players

**DOI:** 10.3390/jfmk6020041

**Published:** 2021-05-08

**Authors:** Senda Sammoud, Raja Bouguezzi, Yassine Negra, Helmi Chaabene

**Affiliations:** 1Research Unit (UR17JS01) Sports Performance, Health & Society, Higher Institute of Sport and Physical Education of Ksar Saîd, University of “La Manouba”, Manouba 2037, Tunisia; senda.sammoud@gmail.com (S.S.); rajabouguezzi@hotmail.com (R.B.); yassinenegra@hotmail.fr (Y.N.); 2Division of Training and Movement Sciences, University of Potsdam, Research Focus Cognition Sciences, D-14469 Potsdam, Germany; 3High Institute of Sports and Physical Education of Kef, University of Jendouba, Jendouba 8189, Tunisia

**Keywords:** change of direction speed, responsiveness, reproducibility, team sport, physical fitness

## Abstract

Background: This study aimed to examine the reliability and sensitivity of a change of direction deficit (CoDD) and to establish its relationship with linear sprint speed. Methods: In total, 89 prepubertal male soccer players participated in this study (age = 11.7 ± 1.2 years, maturity offset = −2.4 ± 1.0). Participants performed the 505 CoD speed test and the 20 m linear sprint speed test with a split interval at 5 m and 10 m. The CoDD was calculated as the mean 505 CoD time—the mean 10 to 20 m time interval. To evaluate the reliability of CoDD, the 505 CoD speed test, and 20 m linear sprint speed were performed twice, one week apart. The sensitivity of CoDD was identified by comparing the values of the typical error of measurement (TEM) and smallest worthwhile change (SWC). Results: Results of the reliability analysis indicated an intraclass correlation coefficient (ICC3.1) < 0.50 (0.47) and a TEM expressed as the coefficient of variation > 5% (10.55%). The sensitivity analysis showed that the ability of the CoDD measure to detect small performance changes is “marginal” (TEM (0.12) > SWC0.2 (0.04)). However, good absolute and relative reliability were observed for the 505 CoD speed test (ICC3.1 = 0.75; TEM < 5%). Alike CoDD, the ability of the 505 CoD speed test to detect small performance changes was rated as “marginal” (TEM (0.07 s) > SWC0.2 (0.04 s)). The CoDD revealed a large association with the 505 CoD speed test (r = 0.71). However, non-significant associations were detected between the CoDD and 5 m, 10 m, and 20 m linear sprint speed intervals (r = 0.10 to 0.16, all *p* > 0.05). Likewise, non-significant correlations between the 505 CoD speed test and 5 m, 10 m, and 20 m linear sprint speed intervals were observed (r = 0.14 to 0.20, all *p* > 0.05). Conclusions: The CoDD displayed poor reliability and limited ability to detect small changes in performance in prepubertal male soccer players. Due to its limited practical utility, practitioners are advised not to consider CoDD scores during the assessment of prepubertal male soccer players.

## 1. Introduction

The ability to change direction rapidly while sprinting is a crucial fitness attribute in soccer [1]. Soccer players’ change of direction (CoD) speed performance is determinant during decisive offensive and defensive actions [2]. In this sense, sports science practitioners, as well as coaches, need to select the appropriate test to evaluate CoD speed of soccer players. However, a major limitation of many of the available CoD speed tests is their long duration. For instance, two of the most commonly used CoD speed tests with soccer players, which are the *t*-test and Illinois CoD speed test, last from 10 to 12 s [3] and 14 to 18 s, respectively [3,4]. This means that not only the ability of the player to rapidly change direction is assessed but also their metabolic capacities [5]. This could harm the content validity of the test. The 505 test appears to be a good alternative to assess deceleration and re-acceleration qualities, alongside the ability to rapidly change direction [6]. The completion of the 505 test takes 2 to 3 s [7,8] which means that the metabolic involvement is reduced compared with other tests (e.g., *t*-test, Illinois CoD speed test). Likewise, due to its relatively short duration, the 505 CoD test may place a greater emphasis on CoD ability compared with other CoD speed tests [8].

Another inherent issue of the available CoD speed tests, including 505, is the substantial contribution of the linear sprint speed capacity to the final performance. For example, Negra et al. [9] revealed very large associations between the *t*-test and 20 m linear sprint speed test (r = 0.85; common variance = 72%) in youth male handball players aged 12 years. For the 505 test, changing direction is reported to account for only 31% of the total 505 times in female softball players [10]. As such, it appears that a large part of the total time to complete the CoD speed tests can be explained by the athlete’s linear sprint speed capacities [8]. This implies that performance in CoD speed tests could be biased by the linear sprinting ability of the player [8,11].

To rule out the influence of linear sprint speed during CoD speed tests, Nimphius et al. [8] suggested a novel metric called CoD deficit (CoDD). The essence of CoDD is to assess CoD speed independent of the linear sprint speed level of the subject [12]. It represents the additional time that a directional change requires when compared to a straight sprint over an equal distance (e.g., 10 m linear sprint speed time versus 505 CoD test time) [12]. As such, CoDD provides insights about the time it takes the athlete to change direction and excludes any effects of linear sprint speed on final CoD speed performance [12].

Reliability is a fundamental characteristic for every physical performance test to be widely applied [13,14]. In particular, reliability tells whether a testing protocol is capable of measuring an outcome variable with consistency, meaning that repeated measurement should provide the same results when the testing conditions are standardized [13,14]. As such, establishing the reliability of a physical performance test is key to make sure that changes in performance are interpreted appropriately. To the best of our knowledge, only Taylor et al. [15] have examined the reliability of the modified 505 and CoDD in a cohort of elite youth soccer players of different age categories (U12 to U18). The same authors revealed that modified 505 CoD speed test and CoDD displayed very low to moderate absolute (typical error of measurement [TEM] ranged from 2.2 to 3.2% and 7.1 to 12% in the modified 505 CoD and CoDD test, respectively) and relative reliability (intraclass correlation coefficient [ICC] ranged from 0.31 to 0.82 and from 0.19 to 0.71 in the modified 505 CoD and CoDD test, respectively). However, the reduced number of prepubertal players (*n* = 33) in the study of Taylor et al. [15] makes the findings far from being conclusive. Therefore, future replication studies involving a larger sample size appear to be needed. In fact, replication studies are fundamental for science to ensure the integrity and validity of findings [16,17]. If a researcher can replicate a study’s results, it means that it is more likely that those results can be generalized to the larger population [16]. Furthermore, there is evidence indicating that faster players displayed longer CoDD [11,18]. In other terms, it seems that players with higher linear sprint speed capabilities are less efficient at changing directions, as assessed via CoDD [11,18,19]. Of note, none of the available studies have examined the association between CoDD and linear sprint speed in prepubertal players, highlighting a void in the literature. Therefore, whether the findings observed in pubertal and post-pubertal subjects apply to prepubertal soccer players is unclear and needs to be further investigated.

Therefore, the aims of this study were i) to examine the test-retest reliability and sensitivity of the 505 CoD speed test and CoDD in a large sample of prepubertal male soccer players and ii) to explore the relationships between CoDD and linear sprint speed performance in the same population. With reference to the previous literature [15], we hypothesized that the CoDD score has low to moderate reliability in prepubertal male soccer players. We additionally hypothesized that faster players would present a longer CoDD [11,18].

## 2. Materials and Methods

### 2.1. Subjects

In total, 89 prepubertal male national level soccer players from the same regional soccer team, who played in several different positions, participated in this study (*n* = 89, age = 11.7 ± 1.2 years, maturity offset= −2.4 ± 1.0). The maturation status of the participants was estimated using the maturity offset method [20] (Table 1). All participants had a background of at least four years of systematic soccer training involving three to five training sessions per week with a soccer match on the weekend throughout the soccer season. Anthropometric characteristics are detailed in Table 1. All participants and their legal representatives were properly informed about all testing procedures, as well as possible benefits and harms related to the study. All procedures were approved by the local Institutional Review Committee of the Higher Institute of Sport and Physical Education, Ksar Said, Tunisia (UR17JS01). Verbal and written informed consent (legal representatives) and assent (children) were obtained after an explanation of the experimental protocol and its potential benefits and harms.

### 2.2. Data Collection

This study was conducted during the first half of the competitive season (October 2018). The first phase of this study aimed to establish the reliability and the sensitivity of both the 505 CoD test and the CoDD scores. During this phase, each athlete completed on two different occasions, seven days apart, the 505 CoD test followed by the linear sprint speed test (i.e., 20 m). On each day, the aforementioned tests were performed in duplicate. The best trial was retained for statistical analyses. A minimum of 3 min of rest was allocated between trials. Two weeks preceding the experimental measurements; two familiarization sessions of all tests were performed. All tests were completed at the same time of day (i.e., 5 to 7 p.m.) in the absence of wind and the same environmental conditions (e.g., 27 to 28 °C for temperature). The participants were instructed to wear the same footwear during all testing sessions.

### 2.3. Change of Direction Speed

The 505 CoD speed test was administered as previously outlined by Draper and Lancaster [21] using an electronic timing system (Microgate, Bolzano, Italy). Players assumed a starting position 10 m from the start line, ran as quickly as possible through the start/finish line, pivoted 180° at the 15 m line, and returned as fast as possible through the start/finish line. The time of the fastest trial was retained. The CoDD for the 505 CoD speed test was calculated via the formula: the mean 505 CoD speed time—the mean 10 to 20 m time interval [1].

### 2.4. Linear Sprint Speed

Linear sprint speed performance was assessed over 5, 10, and 20 m using an electronic timing system (Microgate, Bolzano, Italy). Participants started in a standing start position 0.3 m before the first infrared photoelectric gate, which was placed 0.75 m above the ground to ensure it captured trunk movement and avoided false signals via limb motion. In total, four single-beam photoelectric gates were used. The ICC for test retest reliability ranged from 0.91 to 0.94. The mean 10- to 20-m split time across these sprints was used to calculate the CoDD for each player [1].

### 2.5. Statistical Analyses

Data are expressed as the mean and standard deviation. Data were tested for normal distribution using the Shapiro–Wilk test. A paired sample *t*-test was applied to determine any learning effect or systematic bias between sample mean scores for test and retest. The effect sizes (ES) was determined according to Cohen’s d and classified as small (0.00 < d < 0.49), medium (0.50 < d < 0.79), and large (d > 0.80) [22]. Relative reliability was determined by calculating the ICC_(3,1)_. We considered an ICC < 0.50 as poor, 0.50 ≤ ICC < 0.75 as moderate, 0.75 ≤ ICC < 0.90 as good, and ICC > 0.90 as excellent [23]. Absolute reliability was analyzed through the TEM expressed as the coefficient of variation (CV). It was calculated by dividing the SD of the difference between scores by √2 [3]. The smallest worthwhile change (SWC) was determined by multiplying the between-subject SD by 0.2 (SWC0.2) [9], which corresponded to a small effect. By comparing SWC with TEM score, test sensitivity in detecting systematic variation in performance can be determined [9]. In the case of TEM < SWC, the test’s capacity to detect small performance changes is considered “good”; when TEM = SWC, it is considered “OK”; when TEM > SWC, it is rated as “marginal” [3]. Pearson’s correlation was used to determine relationships between the CoDD, 505 CoD test, and sprint speed performances. Hopkins [24] has suggested that an absolute correlation coefficient of 0 to 0.1 is considered “trivial,” between 0.11 to 0.33 “small”, between 0.31 to 0.5 “moderate”, between 0.51 to 0.7 “large”, between 0.71 to 0.9 “very large”, 0.9 to 0.99 “nearly perfect”, and 1 “perfect”. The level of significance for all tests was set at *p* < 0.05. All statistical analyses were conducted using SPSS v.24.0 for Windows (SPSS, Inc, Chicago, IL, USA).

## 3. Results

### 3.1. Reliability and Sensitivity Outcomes

Data outcomes for the 505 CoD test and CoDD test are displayed in Table 2.

Our statistical calculations revealed a good relative and absolute reliability for the 505 CoD test (ICC_(3,1)_ = 0.75, and TEM < 5%). In terms of the CoDD scores, the statistical calculations revealed a poor relative and a low absolute reliability (ICC_(3,1)_ = 0.47, and TEM > 5%).

Regarding the sensitivity analysis, the TEM (0.07 s) was higher than the SWC0.2 (0.04 s), indicating a “marginal” ability of the 505 CoD speed test to detect small performance changes. Likewise, the CoDD test could not detect small performance changes (SWC0.2 (0.04 s) < TEM (0.12 s)) (Table 2).

### 3.2. Relationship between CoDD Scores, CoD Speed, and Linear Sprint Speed

A strong relationship between CoDD scores and the 505 CoD test was detected (r = 0.71; *p* < 0.01). However, our statistical calculations showed non-significant associations between the CoDD test and 5 m, 10 m, and 20 m linear sprint speed intervals (r = 0.10 to 0.16, all *p* > 0.05) (Table 3). In addition, non-significant correlations between the 505 CoD test and 5 m, 10 m, and 20 m linear sprint speed intervals were observed (r = 0.14 to 0.20, all *p* > 0.05).

## 4. Discussion

The current study sought to assess the test-retest reliability and sensitivity of CoDD scores and to examine its relationships with linear sprint speed performance in prepubertal male soccer players. The main findings indicated that i) the CoDD score demonstrated poor relative and low absolute reliability, ii) the 505 CoD test presented good reliability but marginal ability to detect small performance change, and iii) no significant association was observed between the CoDD score and linear sprint speed performance in prepubertal male soccer players.

The findings of this study showed poor relative (ICC_(3,1)_ = 0.47) and low absolute reliability (TEM > 5%) of the CoDD score in prepubertal male soccer players. This is in line with the study of Taylor et al. [15], who reported low to moderate relative (ICC ≤ 0.5) and low absolute reliability (TEM = 9%) of the CoDD in prepubertal male soccer players aged 12 years. In addition, and based on the sensitivity analysis, the ability of the CoDD to detect small changes in performance can be rated as “marginal”. Taken together, it seems that the CoDD score is not reliable. Therefore, it is recommended not to use CoDD for the sake of performance monitoring in prepubertal soccer players.

In terms of the 505 CoD speed test, findings indicated a good relative and absolute reliability (ICC_(3,1)_ = 0.75, and TEM < 5%). This observation is per an earlier published study that found a moderate relative and absolute reliability of the modified 505 CoD speed test in prepubertal male soccer players [15]. In addition, a recent study by Kerdaoui et al. [25] indicated high relative and good absolute reliability for the 505 CoD speed test (ICC = 0.80; TEM = 1.66%) in healthy physical education students. It is worth noting that all participants of this study had a background of at least four years of systematic soccer training including three to five training sessions per week and a soccer match on the weekend throughout the soccer season. As such, participants were frequently exposed to CoD speed exercises. This could make them generate stable CoD skills during the test. However, based on the sensitivity analysis, the ability of the 505 CoD speed test to detect small performance changes can be rated as “marginal”.

Results of the correlation analyses revealed a large association between the 505 CoD performance and the CoDD scores (r = 0.71, *p* < 0.01, common variance = 51.5%). Our results are in line with those established by Nimphius et al. [8] that found a very large correlation (r = 0.74 to 0.81) between the 505 CoD test and the CoDD score measured for both legs in male cricketers aged 24 years. Such results may confirm that both tests can measure CoD ability. Regarding the CoDD and linear sprint speed performances, our findings reported non-significant correlations (r = 0.107 to 0.161; all *p* > 0.05). Nimphius et al. [8] revealed statistically non-significant correlations (r = −0.08 to 0.09) between 10 m and 30 m linear sprint speed times and the CoDD scores measured for both legs in male cricketers aged 24 years. Likewise, Nimphius et al. [26] showed a small and non-significant relationship (r = 0.19) between CoDD and linear sprint speed time in Collegiate Division I American football players aged 18 to 22 years. The present as well as earlier studies [8,26] suggested that the CoDD provides a practical way to remove the influence of the linear sprint speed on such tests of CoD speed. More specifically, the non-significant correlation between CoDD and linear sprint speed time indicates that CoDD represents a unique measure of physical performance.

Some methodological limitations related to this study warrant discussion. For instance, strength and biomechanical tests (e.g., maximal dynamic strength, rate of force development, vertical/horizontal jump tests) should be implemented in future studies to obtain in-depth knowledge regarding their associations with the CoDD test. Likewise, it has to be emphasized that correlations do not establish cause-and-effect relations. Cross-sectional (i.e., correlational) studies simply show the magnitude of the interrelation between two variables. In other words, the associations between linear sprint speed time and CoDD performances do not accurately reflect cause-and-effect relations. Therefore, the correlation between variables reported in this study should be interpreted with caution.

## 5. Conclusions

In agreement with an earlier study [15], the findings of the present study indicated poor reliability of the CoDD in prepubertal male soccer players. Additionally, the ability of the CoDD to detect small performance changes was “marginal”. However, the 505 CoD test presented good reliability but, as with CoDD, “marginal” ability to detect small changes in performance. No significant association was observed between the CoDD and linear sprint speed. Therefore, practitioners should be aware that CoDD presented limited practical utility in prepubertal male soccer players.

## Figures and Tables

**Table 1 jfmk-06-00041-t001:** Characteristics of the subjects.

	Subjects (*n* = 89)
Age (years)	11.7 ± 1.2
Height (cm)	149.6 ± 9.9
Body mass (kg)	38.4 ± 7.4
Maturity offset	−2.4 ± 1.0
APHV (years)	14.1 ± 0.6

APHV: age at peak height velocity.

**Table 2 jfmk-06-00041-t002:** Intraclass correlation coefficient, typical error of measurement, and smallest worthwhile change for 505 CoD speed test and change of direction deficit in prepubertal male soccer players.

	Variables	Test	Retest	*p*	ICC_(1,3)_ [95%CI]	TEM	TEM%	SWC_0.2_
**Participants** (***n* = 89**)	505 CoD test (s)	2.68 ± 0.18	2.69 ± 0.17	0.66	0.75 [0.62–0.83]	0.07	2.59	0.04
	CoDD scores (s)	1.13 ± 0.22	1.11 ± 0.16	0.44	0.47 [0.19–0.65]	0.12	10.55	0.04

ICC: Intraclass correlation coefficient; CI: Confidence interval; TEM: Typical error of measurement; SWC: smallest worthwhile change; CoDD: change of direction deficit.

**Table 3 jfmk-06-00041-t003:** Correlation coefficients between 505 change of direction speed test, change of direction deficit, and linear sprint speed in prepubertal male soccer players.

	505 CoD Speed Test (s)	5 m	10 m	20 m
CoDD (s)				
r	0.71	0.16	0.10	0.13
p	<0.01	0.12	0.31	0.19
505 CoD speed (s)				
r		0.20	0.14	0.20
p		0.053	0.163	0.05

CoDD: change of direction deficit; CoD: change of direction.

## Data Availability

All underlying data are in the text, tables and figures.

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
