# Peer review of "The Reliability and Sensitivity of Change of Direction Deficit and Its Association with Linear Sprint Speed in Prepubertal Male Soccer Players"

_jfmk, 2021, doi:10.3390/jfmk6020041_

Round 1

Reviewer 1 Report

Thank you for having the opportunity to review this paper. the manuscript is well written as is the methodology. Unfortunately, there is no substantial rationale to legitimize the research and a critical discussion of the study's own results.

This paper would enrich the introduction section

Quoting: Dugdale JH, Sanders D, Hunter AM. Reliability of Change of Direction and Agility Assessments in Youth Soccer Players. Sports. 2020; 8(4):51. https://doi.org/10.3390/sports8040051

“” When assessing COD performance, versions of the “505” test are most commonly selected due to their ability to challenge deceleration and reacceleration qualities, alongside providing comparative data between turn legs “”

Sentence 85 is without a reference and seems forced. According to Loturco et al. , an increase in sprint speed with the specialization and corresponding decrease in COD has already been described

Quoting: Loturco, I., Jeffreys, I., Abad, C. C. C., Kobal, R., Zanetti, V., Pereira, L. A., & Nimphius, S. (2019). Change-of-direction, speed and jump performance in soccer players: a comparison across different age-categories. Journal of Sports Sciences, 1–7. doi:10.1080/02640414.2019.1574276 

sprint speed and COD ability appear to remain stable or even decrease across the age-groups. Overall, the linear speed at longer distances (i.e. 10- and 20-m) increases progressively during the specialization process; (…) and (3) surprisingly, the COD deficit presents a gradual increase, as age and level of specialization increase.

82 You cannot have the sole objective of re-evaluating a little-analyzed section of a previous article. In this part of the introduction, the rationale for which it was necessary to conduct this study must be argued.

291 How do you distinguish greater preparation and competitiveness in U13 teams?

Limitations in the manuscript not of the correlation study need to be examined

The conclusions are satisfactory but at times they do not seem to be drawn from the results and from the discussion of the findings.

Author Response

We would like to thank you for your careful perusal of our paper and for the helpful/constructive comments. We have considered all your suggestions in a point-by-point responses. We hope that the paper now complies with the high quality standard of Journal of Functional Morphology and Kinesiology.

Best regards,

Authors

Reviewer 1 (changes in the manuscript are highlighted in green)

Comment 1

Thank you for having the opportunity to review this paper. The manuscript is well written as is the methodology. Unfortunately, there is no substantial rationale to legitimize the research and a critical discussion of the study's own results.

Authors’ reply: Thank you for your positive feedback. We have revised the rationale to make it clearer to the reader. In brief, this study was based on two main aspects. First, to verify the findings of Taylor et al. (2018) having a larger sample size. In other words, this study is a sort of replication of Taylor’s study. In fact, replication studies are fundamental for science to ensure the integrity and validity of the findings (National Academies of Sciences et al., 2019; Peng & Hicks, 2021). If a researcher can replicate a study’s results, it means that it is more likely that those results can be generalized to the larger population (National Academies of Sciences et al., 2019). Second, none of the available studies has examined the association between the change of direction deficit and linear sprint speed in prepubertal players. As such, this work aimed to fill this gap in the literature and to confirm the findings of an earlier study (Taylor et al. 2018) but with larger sample size. The following sentences were embedded in the introduction.

“However, the reduced number of prepubertal players (n=33) in the study of Taylor et al. [14] makes the findings far from being conclusive. Therefore, future replication studies involving a larger sample size appear to be needed. In fact, replication studies are fundamental for science to ensure the integrity and validity of findings [16, 17]. If a researcher can replicate a study’s results, it means that it is more likely that those results can be generalized to the larger population [16].”

“Therefore, the aims of this study were i) to re-examine the test-retest reliability and sensitivity of the 505 CoD speed test and CoDD in a large sample of prepubertal male soccer players and ii) to explore the relationships between CoDD and linear sprint-speed performance.”

Comment 2

This paper would enrich the introduction section

Quoting: Dugdale JH, Sanders D, Hunter AM. Reliability of Change of Direction and Agility Assessments in Youth Soccer Players. Sports2020; 8(4):51. https://doi.org/10.3390/sports8040051

“” When assessing COD performance, versions of the “505” test are most commonly selected due to their ability to challenge deceleration and reacceleration qualities, alongside providing comparative data between turn legs “”

Authors’ reply: Thank you for this. We have considered the suggested reference. The following changes were included in the introduction:

“The 505 test appears to be a good alternative to assess deceleration and re-acceleration qualities, alongside the ability to rapidly change direction [6]. The completion of the 505 test takes 2-3 sec [7,8] which means that the metabolic involvement is reduced compared with other tests (e.g., T-test, Illinois CoD speed test). Likewise, due to its relatively short duration, the 505 CoD test may place a greater emphasis on CoD ability compared with other CoD speed tests [8].”

Comment 3

Sentence 85 is without a reference and seems forced. According to Loturco et al., an increase in sprint speed with the specialization and corresponding decrease in COD has already been described

Quoting: Loturco, I., Jeffreys, I., Abad, C. C. C., Kobal, R., Zanetti, V., Pereira, L. A., & Nimphius, S. (2019). Change-of-direction, speed and jump performance in soccer players: a comparison across different age-categories. Journal of Sports Sciences, 1–7. doi:10.1080/02640414.2019.1574276 

Sprint speed and COD ability appear to remain stable or even decrease across the age-groups. Overall, the linear speed at longer distances (i.e. 10- and 20-m) increases progressively during the specialization process; (…) and (3) surprisingly, the COD deficit presents a gradual increase, as age and level of specialization increase.

Authors’ reply: Thank you for your comment and suggestion. We have included references that substantiate the statement. Please, note that several studies have shown that sprint speed is inversely related to change of direction deficit (Dos’Santos et al. 2019; Freitas et al. 2019; Loturco et al. 2018; Nimphius et al, 2013).  This means that the faster the athlete, the less efficient his/her change of direction deficit. Unlike the study of Loturco et al. (2020), the focus in this study was on one age group (i.e., prepubertal male soccer players). The point is to verify if the inverse relation between linear sprint speed and change of direction deficit observed in earlier studies with adult athletes (Dos’Santos et al. 2019; Freitas et al. 2019; Loturco et al. 2018; Nimphius et al, 2013) would also be seen in prepubertal soccer players. Of note, this wasn’t examined before and this study deals which such a void in the literature. We revised the sentence and now it reads:

“In other terms, it seems that players with a higher linear sprint speed capabilities are less efficient at changing directions assessed via CoDD [11, 18].”

Comment 4

82 You cannot have the sole objective of re-evaluating a little-analyzed section of a previous article. In this part of the introduction, the rationale for which it was necessary to conduct this study must be argued.

Authors’ reply: Thank you for this comment. We would like to point out that this study has two objectives. The first is to replicate a previous one (i.e., Taylor et al. 2018) to ensure the accuracy and validity of the findings. For more details, please be referred to our answer to comment 1. Second, to investigate the relationship between linear sprint speed and change of direction deficit which is an aspect not yet explored. We have included changes in the introduction as follows:

“However, the reduced number of prepubertal players (n=33) in the study of Taylor et al. [14] makes the findings far from being conclusive. Therefore, future replication studies involving a larger sample size appear to be needed. In fact, replication studies are fundamental for science to ensure the integrity and validity of findings [16, 17].  If a researcher can replicate a study’s results, it means that it is more likely that those results can be generalized to the larger population [16].”

Additionally, we revised the first objective of the study as follows:

“Therefore, the aims of this study were i) to re-examine the test-retest reliability and sensitivity of the 505 CoD speed test and CoDD in a large sample of prepubertal male soccer players and ii) to explore the relationships between CoDD and linear sprint-speed performance.”

Comment 5

 291 How do you distinguish greater preparation and competitiveness in U13 teams?

Authors’ reply: Thank you for your comment. Participants of this study have a background of a minimum of 4 years of soccer practice, including four to five training sessions per week with a soccer match on the weekend. We would say that good preparation contributes to competitiveness. However, the idea in the context of this study was not to address what would differ between preparation and competitiveness but rather to explain what would be the reason behind the good test-retest of the 505 CoD speed test in prepubertal soccer players. To make the idea clearer to the reader, we revised the respective sentence as follows:

It is worth noting that all participants of this study had a background of at least four years of systematic soccer training including three to five training sessions per week and a soccer match on the weekend throughout the soccer season. As such, participants were frequently exposed to CoD exercises. This could make them generating stable CoD skills during the test. »  

Comment 6

 Limitations in the manuscript not of the correlation study need to be examined

Authors’ reply: Thank you for your comment. We tried to mention the most prominent limitations of the study. The fact the correlation analysis does not necessarily reflect a cause-and-effect relation is an inherent limitation that should, from our perspective, be mentioned to help to get a better understanding of the findings. However, we would highly appreciate it if the reviewer specifies what further limitations could be mentioned. 

Comment: The conclusions are satisfactory but at times they do not seem to be drawn from the results and from the discussion of the findings.

Authors’ reply: Thank you for the affirmative comment. We revised the conclusion to ensure that it speaks of the main findings.

References used for the revision

Dos’ Santos, T; Thomas, C; Jones, PA; Comfort, P. Assessing asymmetries in change of direction speed performance; application of change of direction deficit. J. Strength Cond. Res. 2019 3: 2953–2961. 

Loturco, I; Nimphius, S; Ronaldo, K; Bottino, A; Zanetti, V; Pereira, L; Jeffreys, I. Change-of direction deficit in elite young soccer players. The limited relationship between conventional speed and power measures and change-of-direction performance. Ger J. Exerc Sport. Res, 2018 

Nimphius, S; Geib, G; Spiteri, T; Carlisle, D. "Change of direction" deficit measurement in division I american football players. J .Aust Strength. Cond. 2013, 21: 115-117. 

Freitas TT, Pereira LA, Alcaraz PE, Arruda AFS, Guerriero A, Azevedo PHSM, Loturco I. Influence of Strength and Power Capacity on Change of Direction Speed and Deficit in Elite Team-Sport Athletes. J Hum Kinet. 2019 Aug 21;68:167-176. doi: 10.2478/hukin-2019-0069. PMID: 31531142; PMCID: PMC6724583.

Taylor JM, Cunningham L, Hood P, et al. The reliability of a modified 505 test and change-of-direction deficit time in elite youth football players. Sci .Med Football. 2018, 3:1–6.

Dugdale, J. H., Sanders, D., & Hunter, A. M. (2020). Reliability of Change of Direction and Agility Assessments in Youth Soccer Players. Sports (Basel), 8(4). doi:10.3390/sports8040051 

National Academies of Sciences, E., Medicine, Policy, Global, A., Committee on Science, E. M., Public, P., . . . Replicability in, S. (2019). In Reproducibility and Replicability in Science. Washington (DC): National Academies Press (US) Copyright 2019 by the National Academy of Sciences. All rights reserved.

Peng, R. D., & Hicks, S. C. (2021). Reproducible Research: A Retrospective. Annu Rev Public Health, 42, 79-93. doi:10.1146/annurev-publhealth-012420-105110

Reviewer 2 Report

I have read with interest this paper where the Authors report data regarding a crucial point of functional evaluations in young athletes, such as reliability and sensitivity of physical performance test.

For such a reason the article should be suitable for publication, nevertheless some minor revisions are needed.

  1. Several times there is a discrepancy in the reported participant’s number:
    • 89 in the abstract at page 1 line 15,
    • 89 and 91 in the text at page 3 lines 100-101,
    • 91 in the table 1 and 2.

Please correct accordingly or clarify if 2 participants were excluded and why.

  1. Methods section:

Page 4 line 119 In the data collection subparagraph, the Authors should explain more in detail if the tests were always performed in this order (1. 505 CoD and 2. Linear speed-sprint) or if the tests were performed randomly following the established interval time of 7 days. This could influence the performance and in turn it could represent a bias in the data analysis. In my opinion it requires a clarification from the Authors.

  1. In the Discussion section Lines 302-304, I suggest to the Authors to add a phrase highlighting even (if present) the originality of their data respect with those reported in Literature (Nimphius et al.).

Author Response

We would like to thank you for your careful perusal of our paper and for the helpful/constructive comments. We have considered all your suggestions in a point-by-point response. We hope that the paper now complies with the high-quality standard of the Journal of Functional Morphology and Kinesiology.

Best regards,

Authors

Reviewer 2 (changes in the manuscript were highlighted in grey)

Comment 1

I have read with interest this paper where the Authors report data regarding a crucial point of functional evaluations in young athletes, such as reliability and sensitivity of physical performance test. For such a reason the article should be suitable for publication, nevertheless some minor revisions are needed.

Authors’ reply: Thank you for your affirmative comment.

Comment 2

 Several times there is a discrepancy in the reported participant’s number:

  • 89 in the abstract at page 1 line 15,
  • 89 and 91 in the text at page 3 lines 100-101,
  • 91 in the table 1 and 2.

Please correct accordingly or clarify if 2 participants were excluded and why.

 Authors’ reply: We apologize for this. The correct number is 89. We have corrected this throughout the revised version.

Comment 3

  1. Methods section:

Page 4 line 119 In the data collection subparagraph, the Authors should explain more in detail if the tests were always performed in this order (1. 505 CoD and 2. Linear svpeed-sprint) or if the tests were performed randomly following the established interval time of 7 days. This could influence the performance and in turn it could represent a bias in the data analysis. In my opinion it requires a clarification from the Authors.

Authors’ reply:  Thank you for your comment. We have clarified the point as follows:  suggested.

During this phase, each athlete completed on two different occasions, seven days apart, the 505 CoD test followed by the linear sprint speed test (e.g., 10-m, and 20-m).”

Comment 4

In the Discussion section Lines 302-304, I suggest to the Authors to add a phrase highlighting even (if present) the originality of their data respect with those reported in Literature (Nimphius et al.)

Authors’ reply: Thank you for your suggestion. Our results confirm those established by Nimphius et al. (2016). Specifically, in her studies, Nimphius et al. (2013) and (2016) included young adults (American football players and cricket athletes, respectively) and revealed statistically non-significant correlations between linear sprint speed times and the change of direction deficit scores. This study included prepubertal male soccer players and showed similar findings as Nimphius et al (2013) and (2016) in that no association was noted between linear sprint speed and change of direction deficit. We revised the respective sentences as follows :

« Nimphius et al. [7] revealed statistically non-significant correlations (r=-0.08 to 0.09) between 10m and 30m linear sprint speed times and the CoDD scores measured for both legs in male cricketers aged 24 years. Likewise, Nimphius et al. [22] showed a small and non-significant relationship (r=0.19) between CoDD and linear sprint speed time in Collegiate Division I American football players aged 18 to 22 years. The present as well as earlier studies [7, 22] suggested that the CoDD provides a practical way to remove the influence of the linear sprint speed on such tests of CoD. More specifically, the non-significant correlation between CoDD and sprint-speed time indicates that CoDD represents a unique measure of physical performance. »

Reviewer 3 Report

I would like to thank the editor for the opportunity to review the manuscript entitled "The reliability and sensitivity of change of direction deficit and its association with sprint speed in prepubertal male soccer players". I find the study design appropriate, and conclusions are support by the results. I have no further comments.

Author Response

Reviewer 3

I would like to thank the editor for the opportunity to review the manuscript entitled "The reliability and sensitivity of change of direction deficit and its association with sprint speed in prepubertal male soccer players". I find the study design appropriate, and conclusions are support by the results. I have no further comments.

Authors’ responses: Thank you for your positive feedback. Highly appreciated.  

Round 2

Reviewer 1 Report

the manuscript has improved considerably, I just suggest the removal from line 96 that "re-"

Thank you

Author Response

The manuscript has improved considerably, I just suggest the removal from line 96 that "re-"

Thank you

Authors' reply: Thank you for the affirmative comment. We made the change as suggested by the reviewer.